# Evaluation of 16S-Based Metagenomic NGS as Diagnostic Tool in Different Types of Culture-Negative Infections

**DOI:** 10.3390/pathogens13090743

**Published:** 2024-08-30

**Authors:** Sara Giordana Rimoldi, Alessandro Tamoni, Alberto Rizzo, Concetta Longobardi, Cristina Pagani, Federica Salari, Caterina Matinato, Chiara Vismara, Gloria Gagliardi, Miriam Cutrera, Maria Rita Gismondo

**Affiliations:** 1Laboratory of Clinical Microbiology, Virology and Bioemergencies, ASST Fatebenefratelli Sacco, Luigi Sacco Hospital, 20157 Milan, Italy; rimoldi.sara@asst-fbf-sacco.it (S.G.R.); tamoni.alessandro@asst-fbf-sacco.it (A.T.); concetta.longobardi@asst-fbf-sacco.it (C.L.); pagani.cristina@asst-fbf-sacco.it (C.P.); federica.salari@unimi.it (F.S.); gloria.gagliardi@unimi.it (G.G.); miriam.cutrera@unimi.it (M.C.); mariarita.gismondo@unimi.it (M.R.G.); 2Microbiology and Virology Laboratory, Fondazione IRCCS Ca’ Granda Ospedale Maggiore Policlinico, 20122 Milan, Italy; caterina.matinato@policlinico.mi.it; 3Clinical Microbiology Laboratory, ASST Grande Ospedale Metropolitano Niguarda, 20162 Milan, Italy; chiara.vismara@ospedaleniguarda.it

**Keywords:** metagenomics, NGS, diagnostics, infection, bacteria, culture-negative sample

## Abstract

Bacterial infections pose significant global health challenges, often underestimated due to difficulties in accurate diagnosis, especially when culture-based diagnostics fail. This study assesses the effectiveness of 16S-based metagenomic next generation sequencing (NGS) for identifying pathogens in culture-negative clinical samples across various medical settings. Overall, 48% of samples were collected from orthopedics, 15% from neurosurgery, and 12% in cardiac surgery, among others. The detection rate of monomicrobial infections was 68.6%, and 5.7% for polymicrobial infections. In addition, NGS detected bacteria in all samples from the lungs, head and neck, and eye specimens. *Cutibacterium acnes* (11%, 12/105) was the most frequent microorganism, followed by *Staphylococcus epidermidis* (10.4%, 11/105), and *Staphylococcus aureus* (9.5%, 10/105). In conclusion, 16S-targeted metagenomic sequencing enhances pathogen detection capabilities, particularly in instances where traditional cultures fail. By the combination of NGS and bacterial cultures, microbiologists might provide a more accurate diagnosis, guiding more effective treatments and potentially reducing healthcare costs associated with empirical treatments.

## 1. Introduction

Bacterial infections represent a significant global health concern [1]. Yet, their impact is often underestimated. The process of isolating microorganisms from patient samples plays a crucial role in diagnosing and managing these infections. It helps identify the causative agent and provides essential information about the organism’s susceptibility to antibiotics, which is vital for effective treatment.

However, traditional culture-based techniques often fall short. They frequently fail to identify the organism and provide subsequent antibiograms. This is particularly true for a range of infectious diseases, including bloodstream infections, endophthalmitis, periprosthetic joint infections, chronic prostatic pelvic syndrome, endocarditis, musculoskeletal infections, infected foot ulcers, meningitis, tuberculosis, hospital-acquired pneumonia, ventilator-associated pneumonia (HAP/VAP), and liver abscesses [2,3,4,5,6,7,8,9,10,11,12]. Specifically, some microorganisms, like anaerobes, are hard to culture and isolate, making conventional procedures time-consuming and costly [13].

A systematic review published in 2022 showed that blood culture-negative endocarditis occurs in about 40% of infective endocarditis, observing that, despite some disadvantages, metagenomic next generation sequencing (NGS) is valuable in pathogen identification [14].

Culture-negative cases represent a major concern in both adult and pediatric sepsis. A retrospective study on 2499 adult patients observed that about 40% of patients with septic shock are culture-negative [15,16]. 

Managing patients suspected of bacterial infection with negative culture results is a significant challenge. The lack of a definitive diagnosis can lead to antibiotic misuse, further expensive analyses, and patient discomfort. This can result in prolonged illness, increased healthcare costs, and a potential rise in antibiotic resistance due to inappropriate antibiotic use.

Recent advancements in metagenomic NGS have introduced new possibilities for detecting multiple pathogens in a single assay. These advancements offer substantial benefits over culture-based techniques and highlight the limitations of traditional microbiological diagnostic methods. The use of NGS platforms in microbiological laboratories is increasing, especially in cases of culture-negative results [17,18].

This study aims to evaluate the effectiveness of 16S-based metagenomic NGS assays in detecting pathogens in clinical samples that have returned negative results in bacterial culture. In doing so, we hope to highlight the potential of NGS as a powerful tool for accurately diagnosing bacterial infections. This could contribute to improved patient outcomes and a reduction in the global health burden caused by these infections. This study may provide valuable data about cases where, despite the failure of traditional cultures to detect bacteria, NGS provides positive results for one or more microorganisms. This could potentially enhance the management of patients with suspected bacterial infections, leading to more targeted and effective treatments.

## 2. Materials and Methods

### 2.1. Standard Culture Methods

In traditional cultures, we incubated all collected samples for both aerobic and anaerobic bacteria on selected agar plates and in brain heart infusion broth. Plates were incubated at 37 °C in a WASPLab automation system (Copan, Brescia, Italy) and observed over a period of 5 days. Isolates were identified at the species level using MALDI-ToF Vitek MS (bioMérieux, Marcy-l'Étoile, France) and were tested for antimicrobial susceptibility with the automated analyser Vitek 2 (bioMérieux, Marcy-l'Étoile, France).

### 2.2. Culture-Negative Samples

From January 2023 to August 2024, culture-negative samples collected from different clinical settings were received at the Laboratory of Clinical Microbiology of a large hospital in Milan, Italy, and tested through NGS. Additionally, we sequenced isolates that could not be identified with traditional laboratory methods.

### 2.3. Next Generation Sequencing

For NGS analysis, we treated the samples according to the manufacturer’s protocol (Invitrogen, Thermo Fisher Scientific, Waltham, MA, USA), and then pooled them for DNA extraction. Tissue/biopsy, abscess, and lymph node samples underwent a pre-treatment of 3 to 12 h depending on their size. DNA purification was performed using the PureLink™ Microbiome DNA Purification Kit (Invitrogen, Thermo Fisher Scientific, Waltham, MA, USA).

Through the Ion 16S™ Metagenomics Kit (Ion Torrent, Thermo Fisher Scientific, Waltham, MA, USA) on Ion Chef System (Ion Torrent, Thermo Fisher Scientific, Waltham, MA, USA) and Ion GeneStudio S5 (Ion Torrent, Thermo Fisher Scientific, Waltham, MA, USA), we performed metagenomic sequencing of the 16S ribosomal RNA region to investigate the seven most conserved hypervariable bacterial regions (primer set V2–4–8, 3–6, and 7–9). Library preparation was performed using the Ion Plus Fragment Library Kit and the Ion Xpress™ Barcode Adapters 1–16 Kit (Ion Torrent, Thermo Fisher Scientific, Waltham, MA, USA).

This allowed us to indicate the taxonomic levels of family, genus, and species. We queried sequences against the Curated Greengenes v13.5 and MicroSEQ ID 16S Reference Library v2013.1 databases. The full 16S kit can identify 107 different taxonomic genera, and the database contains all deposited bacterial genomes. Based on past laboratory experience, and considering the numerous bacterial species found within a biological sample, including the non-pathogenic bacterial microbiota, we set a cut-off of 1000 counts for frequency of reads (FDR) to interpret the isolated bacteria as a potential pathogen. FDR is considered the average number of reads (fragments of sequence) that align with or “cover” a known reference base to achieve bacterial identification [19,20].

## 3. Results

A total of 105 samples collected from patients receiving empiric antibiotic therapy were analyzed by means of NGS: 48% (50/105) were collected in orthopedics wards, 15% (16/105) in neurosurgery, 12% (13/105) in cardiac surgery, and 10% (11/105) in general surgery. In addition, NGS was performed on 13 isolates unidentifiable with traditional laboratory methods. The detailed list of samples is reported in Table 1.

Among culture-negative samples, 74.3% (78/105) tested positive to one or more microorganisms through NGS. The positivity rate by sample type was as follows: 80% (40/50) for orthopedics, 62.5% (10/16) for CNS, 69.2 (9/13) for heart, 63.6% (7/11) for general surgery, 25% (1/4) for fluids, and 100% for lung (6/6), head and neck (H&N) (3/3), and eye (2/2) samples.

The detection rate of single pathogens was 68.6% (72/105), while two or more pathogens were detected in 5.7% (6/105) of samples. Specifically, 32 different bacteria were detected: *Cutibacterium acnes* (11%, 12/105) was the most frequent, followed by *Staphylococcus epidermidis* (10.4%, 11/105) and *Staphylococcus aureus* (9.5%, 10/105). The proportion of samples negative for both culture and NGS analyses was equal to 25.7% (27/105). Table 2 shows all the microorganisms observed through NGS analysis.

The frequency of microorganisms varied among the different clinical settings and sample types: *S. epidermidis* (20%, 10/50) was the most frequent for orthopedics, a polymicrobial infection of *Prevotella* spp.—*C. acnes* (13%, 2/16) for CNS, *S. gallolyticus* (15%, 2/13) for the heart, and *S. aureus* (18%, 2/11) for general surgery. Of the samples collected from the lung (6/6), H&N (3/3), and eye (2/2), 100% showed evidence of NGS-detectable bacteria. On the other hand, the highest frequency, 37.5% (6/16), of culture-negative and NGS-negative samples was observed in samples collected from CNS. The complete NGS results are reported in Table 3.

Considering isolates not identifiable using traditional laboratory methods, NGS analysis revealed nine different bacteria. Specifically, *A. lwoffi* and *Peptostreptococcus* spp. were the most frequent microorganisms, with a frequency of 15% (2/13) each, see Table 4.

## 4. Discussion

The results of this study underscore the significant potential of next generation sequencing (NGS) in identifying bacterial pathogens in samples that yield negative results in traditional culture methods. Our findings demonstrate that targeted metagenomic NGS, specifically 16S-based, can detect a diverse array of bacterial species, many of which are not detectable through conventional techniques. This highlights the importance of integrating advanced molecular diagnostics into routine clinical practice to enhance pathogen detection and patient management.

One of the key observations from our study is the high detection rate of bacteria using NGS in various clinical samples, particularly in orthopedics, neurosurgery, and cardiac surgery settings. This aligns with previous studies indicating the utility of NGS in diagnosing infections where traditional cultures fail, such as in cases of endophthalmitis and periprosthetic joint infections [3,4,21]. The ability of NGS to provide comprehensive microbial profiles can lead to more accurate and timely diagnoses, which are crucial for initiating appropriate treatments and improving patient outcomes. Our results are consistent with other studies that managed to obtain valuable information for the diagnosis and treatment of culture-negative patients suspected of having bacterial infection [22]. In our study, we observed an NGS positivity rate equal to 74.3%, which is consistent with another study that reported that up to 30% of culture-negative samples might test negative at NGS analysis too [23]. 

Moreover, the identification of bacteria like *A. lwoffii* and *Peptostreptococcus* spp. suggests that NGS can play a vital role in identifying atypical or fastidious pathogens that are often missed by traditional methods. This could be particularly beneficial in managing complex infections and reducing the burden of antibiotic resistance by promoting targeted antimicrobial therapy [15]. On the other hand, NGS-based methods overcome one important limitation of culture: the isolation and identification of anaerobic bacteria, which remain difficult to grow [23].

In addition, the administration of antibiotics before surgical procedures might impact the results of bacterial culture in patients suspected with post-operative infection, highlighting the importance of using NGS in combination with bacterial culture [24].

Despite its advantages, the application of NGS in clinical microbiology is not without challenges. The interpretation of NGS data requires careful consideration of the clinical context to distinguish between true pathogens and contaminants or commensal organisms. Additionally, the high sensitivity of NGS can sometimes result in the detection of multiple organisms, complicating the clinical decision-making process [25]. Thus, developing standardized guidelines for the interpretation of NGS results is essential to maximize its diagnostic utility. On the other hand, for a single identification, conventional methods are less expensive. A CDC analysis estimated a cost of 150–200 US dollars per bacterial isolate. The cost, along with the need of skilled operators and high-end technologies, represent other factors that prevent the widespread implementation of NGS [26].

Furthermore, our study indicates that while NGS is a powerful tool, it should complement rather than replace traditional culture methods. Traditional cultures remain valuable for antimicrobial susceptibility testing, which is crucial for guiding effective treatment regimens. However, it should be noted that in some culture-negative cases, genome sequencing provided an identification and guided rational antimicrobial therapy as well [22,27].

Integrating NGS with culture-based methods can provide a more comprehensive diagnostic approach, combining the strengths of both technologies to improve infection diagnosis and management. As reported by Ma et al., metagenomic NGS might detect microorganisms with higher sensitivity and with a shorter turnaround time, despite the use of antibiotics before sampling. In addition, NGS might reduce the misuse of antibiotics [28]. As consequence, NGS might be deployed in critical situations where patients are strongly suspected of having a bacterial infection, but where conventional methods test negative or require too much time.

While this study presents significant findings, several limitations must be acknowledged: (1) sample size: the limited number of samples does not fully represent the broad spectrum of clinical infections, (2) the lack of clinical correlation: correlating NGS findings with patient symptoms, treatment responses, and clinical outcomes would provide a more robust assessment of its diagnostic value, and (3) the standardization of data interpretation: there is a need for standardized protocols and interpretation guidelines to ensure consistency and reliability across different laboratories.

Addressing these limitations in future research will be crucial for optimizing the clinical utility of NGS and ensuring its integration into routine diagnostic workflows.

Our findings demonstrate the enhanced diagnostic capabilities of NGS for identifying bacterial pathogens in culture-negative clinical samples. The integration of NGS into clinical microbiology can lead to improved diagnostic accuracy and patient care, underscoring the need for its broader adoption in routine diagnostics.

## 5. Conclusions

In conclusion, this study highlights the significant benefits of incorporating NGS into routine clinical diagnostics, particularly for culture-negative infections. By providing a broader and more accurate detection of pathogens, NGS has the potential to improve patient outcomes, reduce healthcare costs, and combat the growing issue of antibiotic resistance. Future research should focus on refining NGS protocols and developing robust clinical guidelines to optimize its application in clinical practice.

## Figures and Tables

**Table 1 pathogens-13-00743-t001:** Number of culture-negative samples collected in different clinical settings and tested with NGS (*n* = 105). In addition, *n* = 13 samples unidentifiable at species level with traditional methods were tested with NGS. Abbreviations: CNS = central nervous system; H&N = head and neck.

Sample Type	Sample Number	
Total Samples (*n*)	105	%
**orthopedics**	**50**	**0.48**
calcaneus biopsy	1	0.01
joint fluid	4	0.04
synovial tissue	3	0.03
bone	40	0.38
articular capsule	2	0.02
**CNS**	**16**	**0.15**
cerebrospinal fluid	5	0.05
ventriculoperitoneal shunt	8	0.07
brain abscess	1	0.01
ventricular biopsy	2	0.02
**heart**	**13**	**0.12**
heart valve	13	0.12
**general surgery**	**11**	**0.10**
breast implant	1	0.01
esophageal biopsy	1	0.01
duodenal biopsy	1	0.01
abdominal resection	1	0.01
wound	2	0.02
gluteal abscess	2	0.02
lymph node	3	0.02
**lung**	**6**	**0.06**
lung biopsy	1	0.01
pleural fluid	1	0.01
bronchial brushing	3	0.03
sputum	1	0.01
**fluids**	**4**	**0.04**
blood culture	2	0.02
dialysis fluid	1	0.01
ascitic fluid	1	0.01
**H&N**	**3**	**0.03**
maxillary sinus	1	0.01
tracheal tissue	1	0.01
auricle bone	1	0.01
**eye**	**2**	**0.02**
ocular prosthesis	1	0.01
vitreous humor	1	0.01
**culture unidentifiable**	**13**	

**Table 2 pathogens-13-00743-t002:** Frequency of detection of bacteria, including single and multiple infections.

	*n*	
Samples	105	%
**Microorganism**		
*Cutibacterium acnes*	12	0.11
*Staphylococcus epidermidis*	11	0.10
*Staphylococcus aureus*	10	0.10
*Streptococcus gallolyticus*	4	0.04
*Finegoldia magna*	4	0.04
*Fusobacterium nucleatum*	3	0.03
*Abiotrophia defectiva*	3	0.03
*Actinomyces* spp.	4	0.04
*Bacteroides vulgatus*	2	0.02
*Corynebacterium* spp.	2	0.02
*Prevotella* spp.	4	0.04
*Staphylococcus intermedius*	2	0.02
*Streptococcus pneumoniae*	3	0.03
*Alloprevotella* spp.	1	0.01
*Berthella aurantiaca*	1	0.01
*Bacteroides fragilis*	1	0.01
*Bacillus cereus sushi*	1	0.01
*Bacteroides* spp.	1	0.01
*Burkholderia* spp.	1	0.01
*Enterobacter cloacae*	1	0.01
*Enterobacter* spp.	1	0.01
*Granulicatella* spp.	1	0.01
*Haemophilus parainfluenzae*	1	0.01
*Klebsiella pneumoniae*	1	0.01
*Leifsonia aquatica*	1	0.01
*Parvimonas micra*	1	0.01
*Paenibacillus* spp.	1	0.01
*Proteus* spp.	1	0.01
*Pseudomonas* spp.	1	0.01
*M. catarrhalis*	1	0.01
*Veillonella* spp.	1	0.01

**Table 3 pathogens-13-00743-t003:** Detection frequency of microorganisms in different clinical settings and materials. Abbreviations: NGS = next generation sequencing; CNS = central nervous system; H&N = head and neck.

	Biological Specimen	NGS Result	*n*	
	**orthopedics—total**		**50**	**%**
**orthopedics**	calcaneus biopsy			
		negative	1	0.02
	joint fluid			
		*C. acnes*	1	0.02
		negative	3	0.06
	synovial tissue			
		*K. pneumoniae*	1	0.02
		*C. acnes*	1	0.02
		negative	1	0.02
	articular capsule			
		negative	2	0.04
	bone			
		*F. nucleatum*	2	0.04
		*Actinomyces* spp.	1	0.02
		*B. vulgatus*	2	0.04
		*Proteus* spp.	1	0.02
		*S. aureus*	6	0.12
		*C. acnes*	6	0.12
		*Prevotella* spp.	1	0.02
		*Corynebacterium* spp.	1	0.02
		*E. cloacae*	1	0.02
		*F. magna*	3	0.06
		*S. epidermidis*	10	0.20
		*P. micra*	1	0.02
		*B. smithii*	1	0.02
		*A. defectiva*	1	0.02
		negative	3	0.06
	**CNS—total**		16	%
**CNS**	cerebrospinal fluid			
		*S. intermedius*	1	0.06
		*Paenibacillus* spp.	1	0.06
		*H. parainfluenzae*	1	0.06
		negative	2	0.13
	ventriculoperitoneal shunt			
		*Pseudomonas* spp.	1	0.06
		*Corynebacterium* spp.	1	0.06
		*C. acnes*	1	0.06
		*K. pneumoniae*	1	0.06
		negative	4	0.25
	brain abscess			
		*S. aureus*	1	0.06
	ventricular biopsy			
		*Prevotella* spp.—*C. acnes*	2	0.13
**heart**	**heart—total**		13	%
	heart valve			
		*S. gallolyticus*	2	0.14
		*A. defectiva*	1	0.08
		*S. pneumoiae*	1	0.08
		*B. fragilis*	1	0.08
		*B. aurantica*	1	0.08
		*Granulicatella*	1	0.08
		*S. gallolyticus*	1	0.08
		*S. intermedius*	1	0.08
		negative	4	0.30
	**general surgery—total**		11	%
**general surgery**	breast implant			
		*Bacteroides* spp.	1	0.09
	esophageal biopsy			
		*F. nucleatum*	1	0.09
	duodenal biopsy			
		negative	1	0.09
	abdominal resection			
		negative	1	0.09
	wound			
		*Enterobacter* spp.	1	0.09
		negative	1	0.09
	gluteal abscess			
		*S. aureus*	2	0.18
	lymph node			
		*S. aureus*—*A. defectiva*	1	0.09
		*Prevotella* spp.	1	0.09
		negative	1	0.09
	**lung—total**		6	%
**lung**	lung biopsy			
		*F. magna*—*Actinomyces* spp.	1	0.17
	pleural fluid			
		*Alloprevotella* spp.	1	0.17
	bronchial brushing			
		*Burkholderia* spp.	1	0.17
		*Veillonella* spp.	1	0.17
		*Actinomyces* spp.	1	0.17
	sputum			
		*S. pneumoniae*—*M. catarrhalis*	1	0.17
**fluid**	**fluid—total**		4	%
	blood culture			
		negative	2	0.50
	dialysis fluid			
		*L. aquatica*	1	0.25
	ascitic fluid			
		negative	1	0.25
**H&N**	**H&N—total**		3	%
	maxillary sinus			
		*Prevotella* spp.—*Actinomyces* spp.	1	0.33
	auricle bone			
		*C. acnes*	1	0.33
	tracheal tissue			
		*C. acnes*	1	0.33
**eye**	**eye—total**		2	%
	ocular prosthesis			
		*S. epidermidis*	1	0.50
	vitreous humor			
		*S. gallolyticus*	1	0.50

**Table 4 pathogens-13-00743-t004:** Frequency of bacteria identified performing 16S-based metagenomic NGS on cultures unidentifiable with traditional laboratory methods. Abbreviations: NGS = next generation sequencing; *n* = number.

	NGS Result	*n*	
Culture Unidentifiable		13	%
	*Acinetobacter lwoffii*	2	0.15
	*Peptostreptococcus* spp.	2	0.15
	*Actinomyces* spp.	1	0.08
	*Bacillus vietnamensis*	1	0.08
	*Enterobacter* spp.	1	0.08
	*Facklamia languida*	1	0.08
	*Pantoea cypripedii*	1	0.08
	*Pseudomonas* spp.	1	0.08
	*Terribacillus* spp.	1	0.08
	negative	2	0.15

## Data Availability

Data are available upon reasonable request to the first author.

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
