# Peer review of "Evaluation of 16S-Based Metagenomic NGS as Diagnostic Tool in Different Types of Culture-Negative Infections"

_pathogens, 2024, doi:10.3390/pathogens13090743_

Round 1
Reviewer 1 Report
Comments and Suggestions for Authors The study addresses next-generation sequencing (NGS) results in specimens that were culture-negative using traditional methods, highlighting the role of NGS as a complementary tool in clinical diagnosis when traditional cultures fail. While the article acknowledges its limitation of not establishing a clinical correlation, this remains a significant shortcoming. To strengthen the manuscript, I recommend the following additions:Traditional cultures often fail due to the patient’s use of antibiotics. Please provide detailed data on the number of patients with NGS-positive (traditional cultures fail) results who did not use any antibiotics throughout their treatment, as well as the number of patients who were on antibiotics at the time of specimen collection. This information would offer valuable insights into the relationship between antibiotic usage and NGS findings. The discussion section notes that NGS cannot fully replace traditional culture methods. It would be beneficial to further clarify under what specific circumstances NGS should be prioritized, and how NGS can provide unique or additional information that could aid in clinical decision-making. Incorporating these details will enhance the manuscript's clinical relevance and scientific rigor. I encourage the authors to consider these revisions.
Author Response
Comment 1: The study addresses next-generation sequencing (NGS) results in specimens that were culture-negative using traditional methods, highlighting the role of NGS as a complementary tool in clinical diagnosis when traditional cultures fail. While the article acknowledges its limitation of not establishing a clinical correlation, this remains a significant shortcoming. To strengthen the manuscript, I recommend the following additions:
Traditional cultures often fail due to the patient’s use of antibiotics. Please provide detailed data on the number of patients with NGS-positive (traditional cultures fail) results who did not use any antibiotics throughout their treatment, as well as the number of patients who were on antibiotics at the time of specimen collection. This information would offer valuable insights into the relationship between antibiotic usage and NGS findings.
Response 1: Thanks for pointing this out. All patients were on empiric antibiotic therapy. We added it in the manuscript and rephrased as follows: "A total of 105 samples collected from patients receiving empiric antibiotic therapy".
Comment 2: The discussion section notes that NGS cannot fully replace traditional culture methods. It would be beneficial to further clarify under what specific circumstances NGS should be prioritized, and how NGS can provide unique or additional information that could aid in clinical decision-making. Incorporating these details will enhance the manuscript's clinical relevance and scientific rigor. I encourage the authors to consider these revisions.
Response 2: Agreed. We added the follwoing statement: "As reported by Ma et al., metagenomic NGS might detect microorganisms with higher sensitivity, with shorter turn-around-time, despite the use of antibiotics before sampling. In addition, NGS might reduce the misuse of antibiotics [20]. As consequence, NGS might be deployed in critical situations where patients are strongly suspected with a bacterial infection but conventional methods result negative or require too much time."
Reviewer 2 Report
Comments and Suggestions for Authors
The article îs of inestimable valye, especially în the case of patients în whom the patogen caudine the infection could not be detected by convenÈ›ional cultures. NGS contributes substanÈ›ialy to a quality diagnostic and terapeutic management, being a real chance for cases of infections, especially periprosthetic with negative cultures. I agree that there îs a need to implement diagnostic and treatment guidelines that include NGS as a complementary diagnostic method în the case of infections with negative cultures
Author Response
Comment 1: The article îs of inestimable valye, especially în the case of patients în whom the patogen caudine the infection could not be detected by convenÈ›ional cultures. NGS contributes substanÈ›ialy to a quality diagnostic and terapeutic management, being a real chance for cases of infections, especially periprosthetic with negative cultures. I agree that there îs a need to implement diagnostic and treatment guidelines that include NGS as a complementary diagnostic method în the case of infections with negative cultures
Response 1: We thank the reviewer for the encouraging comment.
Reviewer 3 Report
Comments and Suggestions for Authors
I read with interest the paper by Rimoldi et al. They applied an NGS assay to a collection of clinical samples with conventional negative culture. The work is simple and straightforward, but it needs to be improved in form and argumentation. Here are some more specific comments.
1. The results seem to be particularly interesting with regard to the diagnosis of anaerobes. The authors should refer to the limitations of conventional diagnostics for this group of bacteria in the introduction and in discussion.
2. There is no reference in the paper to the cost of NGS and the need for a cost-benefit analysis, factors that currently prevent its widespread implementation in microbiology laboratories.
3. Lines 45-46: Empirical antibiotic treatment has to do with clinical signs of infection and not with the identification of bacterial species.
4. Lines 105-106: More appropriate referring to detection rate and not to monomicrobial/polymicrobial infections.
5. The introduction and discussion are not very engaging, there are few bibliographical references and few comparisons with data already published in the literature. They need to be improved.
Author Response
Comment 1: I read with interest the paper by Rimoldi et al. They applied an NGS assay to a collection of clinical samples with conventional negative culture. The work is simple and straightforward, but it needs to be improved in form and argumentation. Here are some more specific comments.
The results seem to be particularly interesting with regard to the diagnosis of anaerobes. The authors should refer to the limitations of conventional diagnostics for this group of bacteria in the introduction and in discussion.
Response 1: We thank the reviewer for pointing this out. We added: in the Introduction section "Specifically, some microorganisms, like anaerobes, are hard to culture and isolate, making conventional procedures time consuming and costly [13]. ", in Discussion "In addition, NGS-based methods overcome one important limitation of culture: isolation and identification of anaerobic bacteria, which remain difficult to grow [19].".
Comment 2: There is no reference in the paper to the cost of NGS and the need for a cost-benefit analysis, factors that currently prevent its widespread implementation in microbiology laboratories.
Response 2: Agreed. We added:”On the other hand, for a single identification, conventional methods are less expensive. A CDC analysis estimated a cost of 150-200 US dollars per bacterial isolate. The cost, along with the need of skilled operators and high-end technologies, represent other factors that prevent the widespread implementation of NGS [22].”.
Comment 3: Lines 45-46: Empirical antibiotic treatment has to do with clinical signs of infection and not with the identification of bacterial species.
Response 3: We rephrased it as follows: “The lack of a definitive diagnosis can lead to antibiotic misuse, further expensive analyses, patient discomfort”.
Comment 4: Lines 105-106: More appropriate referring to detection rate and not to monomicrobial/polymicrobial infections.
Response 4: We provided the overall detection rate and by sample type: "Among culture-negative samples, 74.3% (78/105) resulted positive to one or more microorganisms through NGS. The positivity rate by sample type was: 80% (40/50) for orthopedics, 62.5% (10/16) for CNS, 69.2 (9/13) for heart, 63.6% (7/11) for general surgery, 25% (1/4) for fluids, 100% for lung (6/6), head and neck (H&N) (3/3), eye (2/2) samples. The detection rate of single pathogen was 68.6% (72/105), while two or more pathogens were detected in 5.7% (6/105) of samples".
Comment 5: The introduction and discussion are not very engaging, there are few bibliographical references and few comparisons with data already published in the literature. They need to be improved.
Response 5: We thank the reviewer. We modified the manuscript accordingly.
Round 2
Reviewer 3 Report
Comments and Suggestions for Authors
The authors addressed all my issues.